# Azygos Lobe: Prevalence of an Anatomical Variant and Its Recognition among Postgraduate Physicians

**DOI:** 10.3390/diagnostics10070470

**Published:** 2020-07-10

**Authors:** Asma’a Al-Mnayyis, Zina Al-Alami, Neveen Altamimi, Khaled Z. Alawneh, Abdelwahab Aleshawi

**Affiliations:** 1Department of Clinical Sciences, Faculty of Medicine, Yarmouk University, Irbid 21163, Jordan; 2Department of Medical Laboratory Sciences, Faculty of Allied Medical Sciences, Al-Ahliyya Amman University, Amman 19328, Jordan; z.alalami@ammanu.edu.jo; 3King Abdullah University Hospital, Irbid 22110, Jordan; tamiminiveen94@gmail.com (N.A.); abdelwahhabjamal@yahoo.com (A.A.); 4Department of Diagnostic Radiology and Nuclear Medicine, Faculty of Medicine, Jordan University of Science and technology, Irbid 22110, Jordan; kzalawneh0@just.edu.jo

**Keywords:** azygos lobe, prevalence, misdiagnosis, computed tomography, Jordan

## Abstract

The right azygos lobe is a rare anatomical variant of the upper lung lobe that can be misdiagnosed as a neoplasm, a lung abscess, or a bulla. The aim of this study was to assess the prevalence of right azygos lobe and to evaluate the ability of postgraduate doctors to correctly identify right azygos lobe. We analyzed a total of 1709 axial thoracic multi-detector computed tomography (CT) images for the presence of an azygos lobe. Additionally, a paper-based survey was distributed among a sample of intern doctors and radiology and surgery residents, asking them to identify the right azygos lobe in a CT image and in an anatomy figure. Results showed that the prevalence of the right azygos lobe in the study sample was 0.88%. Men have more right azygos lobes than women. None of the intern doctors or surgery residents identified the right azygos lobe correctly, whereas more than half (57.1%) of the radiology residents did. Most of the incorrect answers about the CT scan were related to the bronchi (25.0%). The apex of the lung (17.7%) and the superior vena cava (17.7%) were the most common incorrect answers about the anatomy figure. In conclusion, the prevalence of the right azygos lobe in the current study is within the range of previously published literature. More education should be given for the identification of the right azygos lobe during anatomy and clinical teaching.

## 1. Introduction

The right azygos lobe (lobus venae azygos) is an anatomical variant present in the upper lobe of the right lung, separated from the rest of the lung by a deep groove called an azygos fissure, which consists of the azygos vein and four layers of pleura (two parietal layers and two visceral layers). The right azygos lobe is formed during the development of the lungs when the precursor to the upper thoracic segment of the azygos vein, called the right posterior cardinal vein, penetrates the apex of the right lung, carrying along the pleural layers instead of arcing forward above the root of the lung as to enter the superior vena cava. The right azygos lobe is detected by conventional chest radiographs and by computed tomography (CT) scans [1,2,3]. It does not correspond to a specific bronchopulmonary segment, and it does not have its own bronchus [4]. The prevalence of the azygos lobe is 1.0% in anatomic specimens, about 0.4% in chest radiographs, and 1.2% in high-resolution CT scans [5,6]. 

Clinicians, anatomists, surgeons, and imaging practitioners should be aware of the possibility of the presence of a right azygos lobe. Since it is usually asymptomatic, its recognition is essential to prevent misdiagnosis, and to be prepared for the technical implications on surgical procedures. It is important to be able to recognize the presence of a right azygos lobe to prevent the wrong interpretation during studying and reading various chest radiologic images, since it might appear as a mass, a lung abscess, or a bulla [5,7]. It could also be misidentified in AIDS patients with tuberculosis, when a fistula between a large lymph node and the esophagus is seen [8]. Furthermore, it might change the normal location of the superior vena cava [9] or cause right paratracheal opacity [5]. The presence of a right azygos lobe might also be associated with other anomalies, such as intrapulmonary right brachiocephalic veins [10] or esophageal atresia [11]. Clinically, surgeons should be ready for different surgical procedures if they encounter the right azygos lobe, since its presence makes surgery using a thoracoscope more difficult, increasing the probability of bleeding throughout the thoracic surgery [12,13] and particularly during sympathectomies [14]. 

As there are few studies about the azygos lobe compared to other anatomical variants, and since many anatomy textbooks and radiology atlases do not mention the right azygos lobe [15], this study will enrich the existing literature with new data about the prevalence of the right azygos lobe in a sample of the Jordanian population. No previous work studied the prevalence of this anatomical variant either in the Jordanian or Arab populations. Moreover, the findings showcase the common misconceptions and potential incorrect answers that could be given by postgraduate doctors upon attempting to identify the right azygos lobe. 

The aims of this study were to assess the prevalence of the right azygos lobe in a sample of the Jordanian population, and to evaluate postgraduate doctors’ ability to identify it correctly. 

The questions to be answered by our study were: (a) is the right azygos lobe present in the Jordanian population, (b) what is its prevalence in the study sample, (c) can postgraduate Jordanian physicians identify the right azygos lobe using CT and anatomy images correctly, and (d) what incorrect answers will the doctors provide when attempting to identify the right azygos lobe?

## 2. Materials and Methods

### 2.1. Prevalence of Right Azygos Lobe

Two of the authors, who are consultant radiologists with more than 10 years of experience in thoracic imaging, studied 1709 axial thoracic multi-detector CT scans of 801 women and 908 men. The sample size was calculated using the Kish formula [16]. For sample size estimation at a 95% significance level with a 5% margin of error, the minimum representative sample size was 386. The anonymous images were chosen randomly from the Picture Archiving and Communication System (PACS) at King Abdullah University Hospital, located in Al-Ramtha in the north of Jordan. A serial number was assigned to each examined scan, and sex and date of birth were also recorded. Every CT scan was analyzed axially for presence or absence of the right azygos lobe. Sagittal and coronal planes were assessed when needed. 

Our inclusion criterium was that the CT scan was for a patient who was recorded as Jordanian according to their identity card, and our exclusion criteria were that the scans were for (a) non-Jordanian patients and (b) patients with pulmonary parenchymal distortion as a result of pulmonary or pleural diseases or due to previous thoracic surgery.

### 2.2. Description of the Right Azygos Lobe

The right azygos lobe is seen in CT images of the right lung using the lung and mediastinal windows. It causes changes in the standard anatomy of the right lung-mediastinal relations. It is identified in the immediate right pre-, para-, and retro-tracheal locations by first identifying the azygos vein and arch (which is shifted superiorly in the case of an azygos lobe) and its drainage in the superior vena cava, with the azygos lobe seen in Figure 1 being inseparable from the postero-medial wall of superior vena cava. The azygos lobe is seen medial to the azygos vein, with its posterior part seen to the right of the esophagus.

### 2.3. Identifying the Right Azygos Lobe Using CT and Anatomy Images

A paper-based evaluating survey was prepared (Figure 2), which included two images of the right azygos lobe: an axial CT image from our current study and an anatomy figure used with permission [17].

This part of the study was performed in a 700-bed teaching hospital in the north of Jordan, which was selected randomly. A random convenient sample of 227 postgraduate physicians was recruited, including 173 intern doctors, which was 56.0% of the total number of intern doctors in the hospital of choice; 26 surgery residents, which was 66.7%; and 28 radiology residents, 90.3%. Intern doctors are postgraduate physicians who must complete a year of postgraduate training (called an internship year) to be able to practice medicine inside Jordan. They were recruited because they are fresh graduates and are expected to have a good background in anatomy. Surgery and radiology residents are trained to be specialists in surgery or radiology, respectively. They should be able to pass the necessary exit exams after completing the 4–5 year long residency period. We recruited them because they are expected to have better knowledge of anatomy and radiology than intern doctors or other residents. 

The evaluating survey was distributed in person and collected by one of the authors, after asking the physicians orally if they agree to answer the provided questions for research purposes. All distributed surveys were evaluated by one of the authors. Answers other than azygos lobe or right azygos lobe were considered incorrect.

Our inclusion criteria were that participants were (a) physicians who were either intern doctors, or surgery or radiology residents at the time of conducting this part of the study (December 2019), and (b) physicians working in the chosen teaching hospital in the north of Jordan. Our exclusion criteria were that they were radiology or surgery specialists. 

The methodologies used in this work fulfill the ethical standards of the Helsinki Declaration. An institutional review board numbered 13/2/3496 was obtained.

## 3. Results 

### 3.1. The Prevalence of the Right Azygos Lobe 

The prevalence of the right azygos lobe in the Jordanian population was found to be 0.88%. The prevalence in men was almost double that in women (Table 1).

### 3.2. Knowledge of Postgraduate Physicians about the Right Azygos Lobe 

None of intern doctors or surgery residents answered the questions identifying the right azygos lobe in CT and anatomy images correctly. However, more than half (57.1%) of the radiology residents identified the right azygos lobe correctly (Table 2).

Most of the incorrect answers identifying the right azygos lobe using the radiological CT image were related to the bronchi (25.0%). The apex of the lung (17.7%) and the superior vena cava (17.7%) were the most common incorrect answers in the anatomy image (Table 3).

## 4. Discussion

The right azygos lobe is a rare anatomical variant in the superior lobe of the right lung. A search for the terms *azygos* and/or *azygous lobe* in the scientific literature revealed that the azygos lobe was never studied in the Jordanian or Arab populations, neither on cadavers nor on any imaging modality. Here, we studied the prevalence of the right azygos lobe in a sample of the Jordanian population and recorded the potential incorrect answers that were provided by a sample of postgraduate doctors when asked to identify the right azygos lobe using anatomy and CT images. 

Rauf et al. (2012) explored the prevalence of the right azygos lobe in various cadaveric populations in their literature review and discovered that it varied from 0.11–1.06%, and from 0.11%–0.2% in radiographic studies [14]. After examining axial CT images, we found that the prevalence of the right azygos lobe in the Jordanian population is 0.88%, which is comparable with the previously reported percentages on cadaveric populations. However, this might be considered a new record because it represents only about half of the previously recorded prevalence in Turkey (1.54%) using thoracic CT scans [18].

We chose CT scans because they provided a clearer and more reliable diagnosis than plain radiographs. Previously, CT scans proved that a group of patients had no pulmonary diseases or mediastinal masses after their X-rays showed abnormal opaque azygos lobes [1]. In patients with pneumothorax, chest X-rays overlook the right azygos lobe [19]. 

The literature also showed that no previous studies were published on the misconceptions about and misidentifications of the right azygos lobe. As our results showed, the structures that the sample of recruited doctors thought was the azygos lobe, both in the CT scan and in the anatomy image, were ambiguous and bizarre. Figure 3 is the same figure used in the survey, fully labeled. 

Table 4 summarizes CT-based descriptions for the structures that were recorded as answers when attempting to identify the azygos lobe.

Regarding the anatomy figure [17], Figure 4 represents the same image used in the survey, fully labeled. A full record of the incorrect answers with our notes and remarks is presented in Table 5.

In this study, we documented a new reading for the prevalence of the right azygos lobe in a sample of Jordanians. The findings can also help anatomy and medicine teaching staff to draw their students’ attention toward various misconceptions about the right azygos lobe, which will improve their anatomical background and their skills in reading CT scans.

The limitations of our study are two-fold: the sample of patients was only from the north of Jordan, and the sample of doctors for the evaluation of postgraduate doctors’ knowledge about the azygos lobe was only from one teaching hospital. Future work might use a more representative sample of the Jordanian population, recruit more doctors to answer the survey, and analyze anatomy and radiology syllabi involving clinically important anatomic variants. We conclude that more attention should be given to the right azygos lobe during anatomy and clinical teaching, since it is present in the Jordanian population.

## Figures and Tables

**Figure 1 diagnostics-10-00470-f001:**
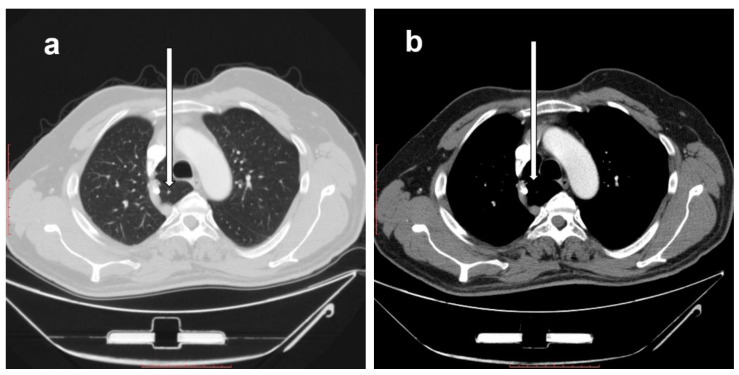
Axial computed tomography (CT) cuts showing the right azygos lobe (white arrows) in the same patient in a lung window (**a**) and in a mediastinal window (**b**). Scale bar = 10 cm.

**Figure 2 diagnostics-10-00470-f002:**
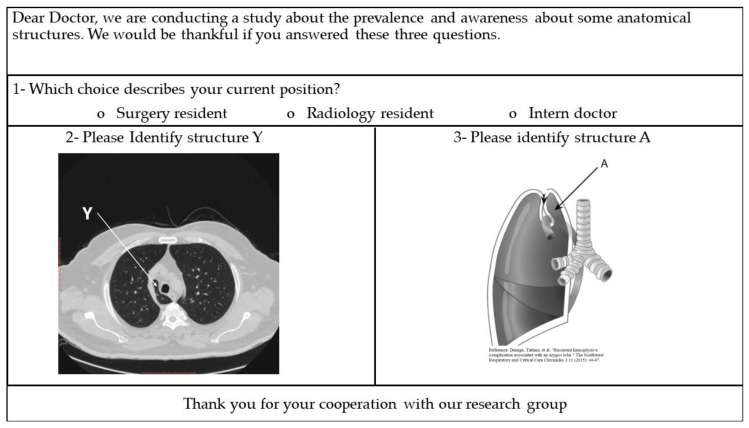
The evaluating survey.

**Figure 3 diagnostics-10-00470-f003:**
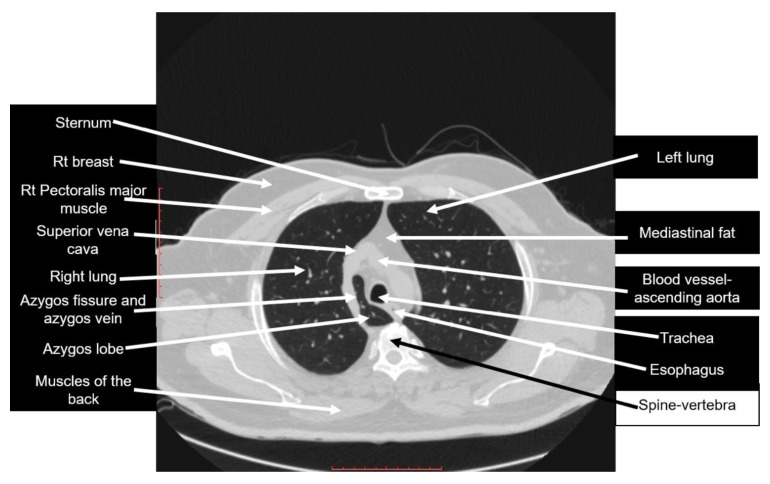
Axial CT scan used in the survey, fully labeled. Scale bar = 10 cm.

**Figure 4 diagnostics-10-00470-f004:**
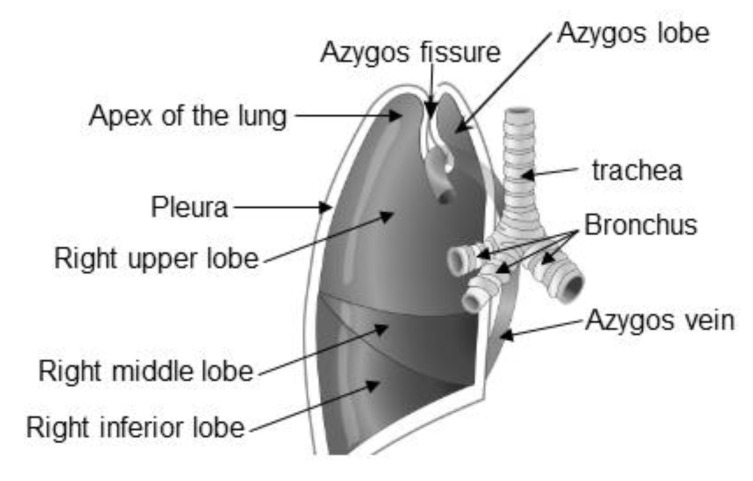
Right azygos lobe used in the survey, fully labeled. (The original figure was adopted after a written permission from [17]).

**Table 1 diagnostics-10-00470-t001:** Prevalence of the right azygos lobe in a sample of the Jordanian population represented as percentages.

	Men	Women	Total
Total number of patients (*n*)	908	801	1709
Cases with an azygos lobe (*n*)	10	5	15
Prevalence (%)	1.01	0.62	0.88

**Table 2 diagnostics-10-00470-t002:** Summary of the results identifying the right azygos lobe represented as numbers and percentages.

Type		Intern Doctors	Surgery Residents	Radiology Residents
Total number of physicians in the teaching hospital (*n*)		309	39	31
Total number of physicians who responded to the survey *n* (%) ^a^		173 (56.0)	26 (66.7)	28 (90.3)
Correct answer *n* (%) ^b^	CT scan	0 (0.0)	0 (0.0)	16 (57.1)
	Anatomy	0 (0.0)	0 (0.0)	16 (57.1)
Incorrect answer *n* (%) ^b^	CT scan	18 (10.4)	12 (46.2)	6 (21.4)
	Anatomy	7 (4.05)	7 (26.9)	3 (10.7)
No answer *n* (%) ^b^	CT scan	155 (89.6)	14 (53.8)	6 (21.4)
	Anatomy	166 (96.0)	19 (73.1)	9 (32.1)

^a^ Percentages are calculated as a proportion of the total number of intern doctors or surgery or radiology residents in the hospital of choice.^ b^ Percentages are calculated as a proportion of the number of intern doctors or surgery or radiology residents who participated in the survey.

**Table 3 diagnostics-10-00470-t003:** A list of incorrect answers identifying the right azygos lobe represented as numbers and percentages. The percentages in the table are calculated as proportions of incorrect answers, not the total number of answers.

Incorrect Answers about the CT Image			Incorrect Answers about the Anatomy Figure		
Structure	*N*	%	Structure	*N*	%
Bronchus/right bronchus/right pulmonary bronchus/right main bronchus	9	25.0	Apex of the lung or right lung apex	3	17.7
Aortic arch	4	11.1	Superior vena cava	3	17.7
Pulmonary trunk	4	11.1	Avascular ring	2	11.8
Superior vena cava	3	8.33	Pleura	1	5.88
Esophagus	3	8.33	Azygous fissure	1	5.88
Inferior vena cava	3	8.33	Apical lung	1	5.88
Trachea	2	5.56	Bronchus/right bronchus/right pulmonary bronchus/right main bronchus	1	5.88
Lung	2	5.56	Aortic arch	1	5.88
Avascular ring	2	5.56	Right upper lobe	1	5.88
Apex of the lung or right lung apex	1	2.78	Lingula	1	5.88
Azygous fissure	1	2.78	Right lung	1	5.88
Apical lung	1	2.78	Circulating aorta	1	5.88
Right upper lobe	1	2.78			
Total	36	100.00		17	100.00

**Table 4 diagnostics-10-00470-t004:** The structures that were provided as incorrect answers when attempting to identify the right azygos lobe using a CT scan, with authors’ remarks.

Structure	CT-Based Description
Bronchus	For its location, please refer to the labeling in Figure 3
Aortic arch	In the provided image, it is not seen as a complete arch, as the level of this image is at its most inferior aspect
Pulmonary trunk	Not seen, as it is just above the level of the provided CT image
Superior vena cava	For its location, please refer to the labeling in Figure 3
Esophagus	A posterior mediastinal structure; for its location, please refer to the labeling in Figure 3
Inferior vena cava	Cannot be in the same image; seen at the level of diaphragm
Trachea	For its location, please refer to the labeling in Figure 3
Lung	This answer is correct, but because it was not specific, it was considered an incorrect answer
Avascular ring	There is no such anatomical term; one possibility could be that the doctor meant a vascular ring, which is a vascular anomaly of the aortic arch system; vascular rings are not present in Figure 3
Apex of the right lung	The apex of the lung is not shown in Figure 3
Azygous fissure	For its location, please refer to the labeling in Figure 3
Apical lung	There is no such anatomical term; it seems that the doctor meant the apex of the lung, which is not shown in Figure 3
Right upper lobe	This answer is correct, but because it was not specific, it was considered an incorrect answer

**Table 5 diagnostics-10-00470-t005:** The structures that were written as incorrect answers when attempting to identify the right azygos lobe using an anatomy figure, with authors’ remarks.

Structure	Anatomy Figure-Based Description
Apex of the lung	For its usual location, please refer to the labeling in Figure 4.
Superior vena cava	The superior vena cava is not shown in Figure 4
Avascular ring	There is no such anatomical term; one possibility could be that the doctor meant a vascular ring, which is the vascular anomaly of the aortic arch system; vascular rings are not shown in Figure 4
Pleura	For its usual location, please refer to the labeling in Figure 4
Azygous fissure	For its usual location, please refer to the labeling in Figure 4
Apical lung	There no such anatomical term; it seems that the doctor meant the apex of the lung, which is labeled in Figure 4
Bronchus	For its usual location, please refer to the labeling in Figure 4
Aortic arch	The aortic arch is not shown in Figure 4
Right upper lobe	This answer is correct, but because it was not specific, it was considered an incorrect answer; please refer to the labeling in Figure 4
Lingula	The ligula is a tongue-like structure near the base of the left lung; Figure 4 is an image of a right lung
Right lung	The answer is correct, but because it was not specific, it was considered an incorrect answer
Circulating aorta	The aorta is not shown in Figure 4

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
