# Peer review of "Azygos Lobe: Prevalence of an Anatomical Variant and Its Recognition among Postgraduate Physicians"

_diagnostics, 2020, doi:10.3390/diagnostics10070470_

Round 1
Reviewer 1 Report
In the manuscript, the authors present a study regarding the incidence of the azygos pulmonary lobe in the northern part of the Jordan and about the ability of the young postgraduate doctors to recognize an azygos lobe on CT-image or an anatomical image. It is an interesting manuscript, but in my opinion this manuscript is not suitable for publication in Diagnostics. My observations are :
- The authors stated that the CT images were evaluated by one consultant radiologist. Usually when you perform a scientific study is preferable that the CT images to be evaluated by at least 2 consultant radiologists.
- The authors stated that they included in the study 229 postgraduate doctors (173 intern doctors, 26 surgery residents and 28 radiology residents). The correct number is 227. So, the statistical analysis is not correct.
- The manuscript does not bring anything new by a scientific point of view. Maybe, this manuscript is more suitable to be publish in a journal that is focusing on the medical education.
Author Response
Dear Editor and reviewer,
- The authors stated that the CT images were evaluated by one consultant radiologist. Usually when you perform a scientific study is preferable that the CT images to be evaluated by at least 2 consultant radiologists.
Thank you for the comment, actually the images were evaluated by 2 consultant radiologists who are both authors of the manuscript; Author 1 (Asma’a Al-Mnayyis) and Author 4 (Khaled Z. Alawneh). It is already mentioned in authors contributions.
This has been edited in the manuscript in line 69 (in materials and methods), highlighted and also tracked
- The authors stated that they included in the study 229 postgraduate doctors (173 intern doctors, 26 surgery residents and 28 radiology residents). The correct number is 227. So, the statistical analysis is not correct.
Thank you for the comment, we apologize for the typing error, yes the correct number is 227 and all our analysis and subanalysis were carried out using this number (227). It is corrected now in the manuscript line 98 (in materials and methods), highlighted and tracked.
- The manuscript does not bring anything new by a scientific point of view. Maybe, this manuscript is more suitable to be publish in a journal that is focusing on the medical education.
Thank you for the comment,
The study has many new points including:
- It is the first study in Jordan and Arab countries to investigate the prevalence of azygos lobe which is a rare anatomic variant. Our findings are expected to be very useful for other researchers from different backgrounds but mainly for anatomists, diagnostic radiologists and surgeons.
- It highlighted how unfamiliar trainees are with anatomy, therefore it is important in bringing attention of the anatomy faculties or staff to focus on such a variant (and other anatomic variants) during teaching in order to characterize such a variant so that it won't be misidentified leading to unnecessary further diagnostic measures or therapeutic interventions.
We did not edit the manuscript for this point because it is already mentioned in discussion starting from line 180.
Reviewer 2 Report
Dear Authors, congratulations. Your paper is smart and interesting, especially when facing the ability to diagnose azygos lobe among young doctors.
It is well written. Statistics are very simple but appropriate.
I have no major suggestions for you. So my opinion is that the paper is worthy for publication.
Good luck!
Author Response
Dear Editor and reviewer,
The reviewer has no major concerns and the reviewer’s points were all positive. We are thankful to his/her input and review.
Reviewer 3 Report
Thank you for allowing me to review the manuscript. I think this is very interesting. An azygous lobe is a phenomenon that we often read about but rarely see. It is interesting to see how unfamiliar trainees are with the anatomy. Very interesting, does not change clinical practice, but nonetheless interesting. I really appreciated the images.
Author Response

(The authors gave the same response as above.)

Reviewer 4 Report
Thank you for submitting this article entitled ‘Azygos Lobe: A Prevalence of an Anatomical Variant and its Recognition among Postgraduate Physicians’.
This article tries to assess the prevalence of azygos lobe in the Jordanian population, and evaluates the skill of young doctors in identifying the azygos lobe on a CT scan.
I think some issues must be assessed: first of all, there is no description of the statistical calculation of the sample size. On a total of 1709 patients, 15 cases of azygos lobe are detected. I think it is not enough to determine the prevalence over the entire population in Jordan. At the same time, it is reductive to perform an analysis on postgraduate doctors in just one hospital. A multicenter survey should be advisable.
With regard to minor points, I would suggest a revision of the article by a native English speaker.
Author Response
Dear Editor and reviewer,
- Reviewer note 1: first of all, there is no description of the statistical calculation of the sample size.
The sample size was calculated using Kish formula. Based on this formula, we assumed that the prevalence in Jordan was unknown since our study is the first to assess the azygos lobe prevalence in the country and in many regional countries. So, using the Kish formula, the required sample was 386 But we studied much more than that, actually we studied 1709 CT samples. Accordingly, we believe that the sample size is correct and more than the required number which represents a strength point for the study.
We clarified this point by adding a paragraph on sample size calculation in materials and methods starting in line 71, with a reference added for the formula (reference 16), and consequently all following references were renumbered. All changes are tracked and highlighted.
- Reviewer note 2: On a total of 1709 patients, 15 cases of azygos lobe are detected. I think it is not enough to determine the prevalence over the entire population in Jordan.
Thank you for the comment, 15 out of 1709 (0.88%) is expected for such rare variant, which has a prevalence that doesn’t exceed 0.11-1.54 in the literature, so it is within the range of what was previously found as mentioned in the introduction (line 39) and the discussion (line 149)
- Reviewer note 3: it is reductive to perform an analysis on postgraduate doctors in just one hospital.
Thank you for the comment, we conducted our study in a large referral hospital with 700 beds capacity. And we have questioned a large percentage of the doctors in concern. We have already mentioned this point in the future work suggestions in our discussion (last paragraph) where we suggested to recruit more doctors to answer the survey in the future studies.
- Reviewer note 4:
Thank you for the comment, we have proofread the article, English had been revised again, all are tracked.
Round 2
Reviewer 1 Report
In the manuscript, the authors present a study regarding the incidence of the azygos lobe in one region from Jordan. The study was conducted in a large hospital from Jordan. The manuscript has been reviewed before and the authors changed the manuscript according to the previous reviewers indications. Their comments are pertinent. That is why I think that this manuscript can be published. Also, I have an observation. Maybe is better to specify in the title of the article that the study was realized in Jordan.
Reviewer 4 Report
Dear colleagues, now that the sample size calculation has been specified, I think the article has gained more value. I think that this article can be of interest for the readers. I renew the invitation to extend the questionnaire also to colleagues who work in secondary or tertiary hospitals